# OpenReview forum: "Breaking the Exploration Bottleneck: Rubric-Scaffolded Reinforcement Learning for General LLM Reasoning"
_ICLR.cc/2026/Conference — Submitted to ICLR 2026_

### Official Review · Reviewer_nBsg · 2025-10-27

**Soundness:** 3
**Presentation:** 3
**Contribution:** 3
**Rating:** 8
**Confidence:** 4

**Summary:**

This paper introduces **Rubric-Scaffolded Reinforcement Learning (RuscaRL)**, a novel instructional scaffolding framework designed to address the persistent **exploration bottleneck** in using Reinforcement Learning (RL) for general Large Language Model (LLM) reasoning tasks. The core problem is that RL improvement requires learning from high-quality samples, but the LLMs' inherent limitations restrict their ability to explore and find new, high-quality samples, leading to an undesirable cycle where what cannot be explored cannot be learned.

RuscaRL leverages **checklist-style rubrics** in two complementary ways:

1.  **Explicit Scaffolding for Exploration:** During the rollout generation phase, rubrics are incorporated as external guidance within task instructions to steer the model towards diverse and high-quality responses. This scaffolding is managed through a two-dimensional control mechanism:
    *   **Intra-Group Scaffolding Differentiation:** Assigns varying levels of rubric criteria to responses within the same sampling group (e.g., using the linear differentiation pattern $\lambda_i = \frac{G-i}{G-1}$) to encourage diversity and guided exploration.
    *   **Inter-Step Scaffolding Decay:** Gradually withdraws the guidance over training steps using a Sigmoid function ($\lambda_{step}(t)$) to encourage the model to internalize the underlying reasoning patterns and minimize reliance on external cues.
2.  **Verifiable Rewards for Exploitation:** Rubrics serve as references for robust LLM-as-a-Judge reward calculation. A Grader LLM performs a binary evaluation ($b_i \in \{0, 1\}$) on each specific criterion $c_i$. The final scalar reward $r_i$ is derived by aggregating these scores and normalizing by the total possible score.

**Key Contributions and Results:** RuscaRL demonstrates superior performance across various benchmarks spanning medical, writing, instruction following, and STEM domains. Notably, RuscaRL significantly boosts the performance of **Qwen2.5-7B-Instruct on HealthBench-500 from 23.6 to 50.3**, surpassing GPT-4.1 (47.9). The fine-tuned Qwen3-30B-A3B-Instruct model achieves 61.1 on HealthBench-500, outperforming leading LLMs like OpenAI-o3 (59.8). Analysis shows RuscaRL improves sampling efficiency, expands the reasoning boundary (at large Best-of-N settings), and generates highly **novel responses** that the initial model could barely generate.

**Strengths:**

**Originality:** The paper pioneers the introduction of **instructional scaffolding theory** (derived from educational psychology, like Vygotsky’s Zone of Proximal Development) into the RLVR paradigm for LLMs. The dual mechanisms of Intra-Group Scaffolding Differentiation and Inter-Step Scaffolding Decay are a highly creative combination designed explicitly to promote high-quality diversity during exploration while ensuring eventual internalization and convergence.

**Quality:** The robustness of the method is confirmed through extensive experimentation across a wide range of tasks and model scales (Qwen, Llama, Instruct, and Base models). The ablation studies rigorously validate the core design choices, showing the **linear differentiation strategy** and the **Sigmoid decay function** are empirically optimal for intra-group and inter-step control, respectively. Furthermore, the analysis of policy entropy demonstrates that RuscaRL achieves a better **exploration-exploitation balance** compared to baselines, avoiding premature entropy collapse or uncontrolled instability.

**Clarity:** The methodology is sharply defined, detailing how rubrics are used both for explicit prompting (scaffolding ratio $\lambda_S$) and for robust reward computation (binary grading and aggregation). The training configuration, hyperparameter settings, and prompt templates (Grader, Scaffolding, Data Generation) are provided in the Appendix, which significantly enhances the study's **reproducibility**.

**Significance:** RuscaRL successfully tackles a fundamental challenge in applying RL to general LLM reasoning: the exploration bottleneck. By enabling smaller, open-source models (like Qwen2.5-7B-Instruct) to match or exceed the performance of much larger or closed-source models (like GPT-4.1 and OpenAI-o3) on difficult benchmarks, the framework demonstrates substantial practical significance for advancing open-source LLM capabilities.

**Weaknesses:**

1.  **Critical Dependency on Rubric Quality and Scarcity:** The authors acknowledge that RuscaRL **critically relies on high-quality, well-structured rubric datasets**, which are currently scarce in the community. The framework is highly sensitive to rubric design quality; poorly designed rubrics (e.g., those with unreasonable point allocations or conflicting criteria) may fail to provide robust reward signals, and narrow rubrics can restrict the generation of diverse, high-quality responses. This dependency poses a practical limitation for applying RuscaRL widely to new domains without substantial prior effort in rubric creation.

2.  **Sensitivity to Decay Hyperparameters:** The crucial Inter-Step Scaffolding Decay mechanism using the Sigmoid function is highly sensitive to the hyperparameters $\alpha$ (steepness) and $t_0$ (midpoint). The ablation study shows that small variations in these parameters can lead to significantly degraded performance (e.g., preventing model adaptation, causing training instability, or inducing overfitting due to over-reliance on external guidance). This suggests that RuscaRL requires meticulous tuning when applied to new tasks or different model architectures to find the optimal decay curve.

3.  **Efficiency and Cost of Reward Calculation:** The reward computation phase necessitates the use of a Grader LLM (e.g., Qwen3-32B or GPT-4.1) for binary evaluation of $N$ criteria per response, across $G$ responses per step. The reported average Reward time of 20 seconds per step, based on external API usage, highlights a potentially significant computational and monetary overhead compared to the Rollout (40 seconds) and Actor Update (15 seconds) stages. The work does not provide a clear estimate of the internal computational cost (e.g., GPU hours, latency) if the Grader LLM were deployed self-hosted, making the framework's overall efficiency profile unclear for self-contained, large-scale training.

4.  **Incomplete Explanation for SFT Superiority in Specific Domains:** While RuscaRL generally outperforms baselines, for the **WritingBench** task (using Qwen2.5-7B-Instruct), Supervised Fine-Tuning (SFT) achieved a larger gain (+17.5) than RuscaRL (+11.0) in the direct setting. Although the authors provide reasoning for the narrowing gap in the SFT-then-RL scenario (overlap in exploration facilitation), a deeper analysis is needed to explain why SFT, based on static GPT-4.1 demonstrations, provided a stronger initial structural foundation for this specific creative task than the dynamic, rubric-based scaffolding provided by RuscaRL.

**Questions:**

1.  **Quantifying Rubric Noise Robustness:** The paper identifies the critical dependence on high-quality rubrics as a limitation. To inform practical application, could the authors conduct an additional analysis quantifying RuscaRL's resilience? Specifically, how does performance change if the rubrics are deliberately perturbed—for example, by inverting points for a fraction of criteria, or by including criteria that are ambiguous or contradictory? This would establish the framework's robustness against real-world "rubric noise."

2.  **Direct Measurement of Scaffolding Internalization:** The inter-step decay mechanism is designed to encourage the model to internalize reasoning patterns. A direct measure of internalization would be to check if the final policy $\pi_{\theta}$ can generate high-quality responses without the explicit rubric cues. Could the authors compare the statistical similarity (e.g., KL divergence or semantic distance) between generations sampled using the final policy *with* scaffolding ($\pi_{\theta}(\cdot|q, R_S)$) versus *without* scaffolding ($\pi_{\theta}(\cdot|q)$)? If internalization is complete, these distributions should be highly similar for high-scoring responses.

3.  **Grader LLM Efficiency and Alternatives for Deployment:** The Reward computation phase adds 20 seconds per step when using external APIs, which is substantial. If the authors used the Qwen3-32B model as the Grader, what is the internal, self-hosted operational cost (e.g., GPU memory usage and latency per evaluation for a typical rubric size)? Furthermore, have the authors explored cost-saving alternatives, such as distilling the knowledge of the Grader LLM into a smaller, faster verifier model, or using a self-training loop to create a low-cost, domain-specific Grader?

4.  **Impact of Maximal Scaffolding on Novelty:** The linear differentiation strategy provides maximal scaffolding ($\lambda_i=1$) to the first sample in each group. While this promotes exploration guided by criteria, does maximum scaffolding ever suppress the emergence of truly novel solutions that do not strictly adhere to the checklist structure? Could the authors analyze whether the highest novelty responses (those with extremely high importance ratios, $\rho_{seq} > 100$) tend to be generated by samples with maximal scaffolding, or instead by those with lower, more moderate scaffolding ratios?

---

> ### Author Response · Authors · 2025-11-21
> **Response (1/4)**
>
> We sincerely appreciate the reviewer for highlighting the originality of our scaffolding framework, the robustness of our experimental validation, and the practical significance demonstrated across diverse models and tasks.
> We have carefully revised the manuscript according to your valuable suggestions.
> Below we address the main points raised in the review.
>
> **[W1]: Critical Dependency on Rubric Quality and Scarcity**
>
> Sorry for the confusion.
> (1) We would like to clarify that our work does **not** focus on **how to generate rubrics**, but rather on **how to use given rubrics more effectively**. RuscaRL provides a methodological contribution rather than a new dataset. The main contribution of RuscaRL is a reinforcement learning framework that integrates rubrics as explicit scaffolding for exploration and as verifiable rewards for exploitation. Using the given rubric datasets, RuscaRL achieves substantial improvements over other rubric-based RL baselines.
>
> (2) It is also worth noting that rubrics have recently emerged as a promising solution for open-ended domains in the research community [1,2,3,4,5,6]. Unlike mathematics or code with objectively verifiable answers, many real-world tasks such as medical consultation and creative writing often require multidimensional evaluation and lack a single, verifiable ground truth. Although rubrics are often well-designed and domain-specific, **these very properties in fact enable them to provide stable and informative signals** for both training and evaluation.
> We view this as a **worthwhile tradeoff**, especially given their broad potential in open-ended applications. For instance, OpenAI introduced HealthBench [1] to evaluate medical dialogue using rubrics, and leading models such as Kimi K2 and Baichuan M2 integrate rubric-based rewards in reinforcement learning for open-ended tasks [2,3]. These trends highlight that the benefits of rubrics outweigh their limitations.
>
> (3) Some recent studies also focus on improving rubric construction itself [5,6,7,8,9,10], and these studies provide valuable advances for building higher-quality evaluation criteria. **Our method is orthogonal to these dataset efforts**, since RuscaRL can integrate with any rubric dataset, regardless of how the rubrics are generated. We hope that our methodological contribution, together with these advances in rubric datasets, will jointly help drive further progress in rubric-based research and promote their effective use in open-ended tasks.
>
>
>
> **[W2]: Sensitivity to Decay Hyperparameters**
>
> Sorry for the confusion.
> While it is true that there exists an **empirically optimal** combination ($\alpha$=125, $t_{0}$=0.2), we would like to emphasize two points:
>
> **(1) RuscaRL outperforms the baseline over a *wide* range of hyperparameters.**
> (i) In Figure 6(c), varying $\alpha$ across a broad range yields a wide plateau where RuscaRL remains significantly above the Rubric-based RL baseline (41.2); only when $\alpha$ is extremely small does the performance drop below the baseline.
> (ii) In Figure 6(d), varying $t_{0}$ across a broad range likewise yields a wide plateau where RuscaRL remains clearly better than Rubric-based RL; only when decay starts excessively early or late do we observe a pronounced degradation in performance.
>
>
> **(2) Practical tuning heuristics are simple.**
> Our experiments suggest the following easy-to-apply guidelines for new tasks or models:
> (i) Choose $\alpha$ in a broad range around 100 to obtain a reasonably steep but not abrupt decay.
> (ii) Choose $t_{0}$ in [0.15, 0.3], so that strong scaffolding dominates early 15–30% of training and then starts to decay.
> Within these ranges, we consistently observe improvements over Rubric-based RL without noticeable instability.

---

> ### Author Response · Authors · 2025-11-21
> **Response (2/4)**
>
> **[W3 & Q3]: Efficiency and Cost of Reward Computation**
>
> Thanks for your insightful comment.
> In our experiments, the policy model (e.g., Qwen2.5-7B-Instruct) is trained on one 8×H200 node, and the Grader model (Qwen3-32B, non-thinking) on an additional node. For each step, we use a batch size of 64 instructions, 8 rollouts per instruction, and an average of 11.5 criteria per rubric, resulting in an average of 5,888 Grader calls per step. The reward stage takes ~60s per step, while the policy computation takes ~40s for rollout and ~15s for update. These costs are comparable to other rubric-based RL baselines using LLMs as judges with multi-criteria scoring (e.g., RaR [4] and Rubicon [5]). Although rubric-based rewards introduce roughly a twofold increase in training cost (a limitation shared by all rubric-based methods rather than specific to ours), we believe this cost is well worth it given the strong performance gains on open-ended tasks.
>
> Moreover, our previous implementation was method-focused and not heavily optimized. We have observed in follow-up runs that significant efficiency gains are possible with relatively simple modifications:
>
> **(1) Lightweight grader models.** We have additionally conducted experiments by replacing Qwen3-32B with the lightweight grader Qwen3-30B-A3B-Instruct-2507. The results in Table R1 show that this modification reduces the per-step reward-stage wall-clock time from 60s to 18s, with only a slight degradation in final performance.
>
> Table R1. Training cost and HealthBench-500 performance with different grader models.
>
> |Grader|Reward Time per training step (s)|Total reward GPU hours|HealthBench-500 score|
> |-|:-:|:-:|:-:|
> |Qwen3-32B (non-thinking)|60|46.7|50.3|
> |Qwen3-30B-A3B-Instruct-2507|18|14.0|48.9|
>
> **(2) Asynchronous rollout-reward strategy.**
> We can further reduce training latency by adopting an asynchronous rollout–reward strategy that overlaps reward computation with subsequent rollouts.
> In the default synchronous pipeline, the per-step latency is:
> $$
> T_{\text{sync}} = T_{\text{rollout}} + T_{\text{reward}} + T_{\text{update}} = 40 + 60 + 15 = 115\text{ s/step}.
> $$
> When the reward stage is run **asynchronously**: each generated sequence is sent to the grader immediately, and grading is overlapped with subsequent rollouts. Then the per-step latency becomes:
> $$
> T_{\text{async}} = \max(T_{\text{rollout}}, T_{\text{reward}}) + T_{\text{update}}.
> $$
>
> Under the same configuration, this reduces the wall-clock time to:
> $$
> T_{\text{async}} = \max(40, 60) + 15 = 75\text{ s/step}.
> $$
>
> With the more efficient Qwen3-30B-A3B grader ($T_{\text{reward}}= 18\text{ s}$), the latency further drops to:
> $$
> T_{\text{async}} = \max(40, 18) + 15 = 55\text{ s/step}.
> $$
> At this point, the latency of the reward stage can be significantly reduced.
>
> *These clarifications have been updated in Appendix C.6 (Pages 23-24) of the revised manuscript.*
>
> **[W4]: Incomplete Explanation for SFT Superiority in Specific Domains**
>
> Sorry for the confusion.
> In the WritingBench setting with Qwen2.5-7B-Instruct, although SFT on GPT-4.1 demonstrations achieves a larger direct improvement than RuscaRL alone, the combined SFT+RuscaRL model still attains the highest overall performance.
> This can be explained by the following factors:
>
> (1) SFT directly fits the full output distribution of GPT-4.1, including content, structure, and subtle stylistic patterns. Since WritingBench is evaluated by LLM judges, this imitation signal provides a particularly strong prior advantage for this benchmark.
>
> (2) For the writing task, the rubrics emphasize topic relevance, structural clarity, coherence, and logical completeness, and are less sensitive to fine-grained stylistic mimicry. RuscaRL therefore mainly improves structural robustness and content coverage, which is useful but not fully aligned with the style-sensitive preferences of the LLM judges.
>
> (3) In the SFT+RuscaRL setting, applying RuscaRL on top of the SFT model further improves WritingBench performance, indicating that SFT and RuscaRL are complementary: SFT supplies a strong style prior, while RuscaRL adds rubric-aligned structure and robustness, leading to the best overall results on this task.

---

> ### Author Response · Authors · 2025-11-21
> **Response (3/4)**
>
> **[Q1]: Quantifying Rubric Noise Robustness**
>
> Thanks for the valuable comment.
> We have additionally conducted experiments to inform practical application by quantifying RuscaRL’s resilience, specifically measuring how performance changes when the rubrics are deliberately perturbed.
> For each rubric, we design the following noise variants:
> - **Original**: the unmodified rubric.
> - **Inverse**: swap high-point and low-point criteria, effectively reversing the relative scoring priorities.
> - **Negated**: flip the sign of each criterion score (e.g., +3 → −3), so "good" behavior is penalized and "bad" behavior is rewarded.
> - **Ambiguous**: inject vague or subjective criteria generated by GPT-4.1.
> - **Contradictory**: inject logically conflicting criteria generated by GPT-4.1.
> - **50% removed**: randomly delete 50% of the original criteria, simulating rubrics with substantially incomplete coverage.
>
> For each noise setting, we train a Rubric-based RL baseline and RuscaRL on medical-domain data using Qwen2.5-7B-Instruct, and evaluate them on four medical benchmarks.
> Table R2 shows that RuscaRL is more robust to rubric noise: under mild perturbations (Ambiguous, Contradictory, 50% removed), it consistently outperforms Rubric-based RL, whereas under severe corruptions (Inverse, Negated) both methods degrade substantially.
> *These additional results have been updated in Appendix C.10 (Pages 25-26) of the revised manuscript.*
>
> Table R2. Robustness to rubric noise on medical benchmarks.
> |Rubric variant|HealthBench-500|LLMEval-Med|MedQA|MedMCQA|
> |-|-|-|-|-|
> |Initial Model|23.4±0.3|48.0±0.3|61.8±0.2|56.3±0.1|
> |+Rubric-based RL|||||
> |├─ Original|41.1±0.1|54.6±0.2|62.1±0.4|56.3±0.1|
> |├─ Inverse|7.1±0.2|41.3±0.5|61.3±0.1|55.8±0.1|
> |├─ Negated|2.9±0.5|36.8±1.4|60.5±0.0|55.6±0.1|
> |├─ Ambiguous|40.1±1.0|54.8±1.2|63.0±0.1|56.2±0.2|
> |├─ Contradictory|43.4±0.7|55.9±1.1|63.3±0.5|55.9±0.2|
> |└─ 50% removed|39.6±1.5|51.7±0.9|62.5±0.6|56.7±0.1|
> |+**RuscaRL (Ours)**|||||
> |├─ Original|**50.3±0.4**|**61.2±0.5**|**63.5±0.1**|**56.5±0.1**|
> |├─ Inverse|10.6±0.7|44.4±0.5|61.3±0.2|56.1±0.1|
> |├─ Negated|6.7±0.5|41.1±0.5|60.8±0.0|55.9±0.0|
> |├─ Ambiguous|46.2±0.3|59.6±1.4|62.2±0.2|56.0±0.1|
> |├─ Contradictory|45.7±0.7|56.8±0.6|63.2±0.5|56.2±0.1|
> |└─ 50% removed|44.6±0.2|54.9±0.6|63.2±0.2|56.3±0.1|
>
>
>
>
> **[Q2]: Direct Measurement of Scaffolding Internalization**
>
> Thanks for the suggestion.
> We have additionally conducted experiments to measure scaffolding internalization in the final policy.
> For each prompt, we compare three responses:
> (i) the initial policy $\pi_{\text{init}}(\cdot \mid q)$,
> (ii) the final policy without scaffolding $\pi_{\text{RuscaRL}}(\cdot \mid q)$, and
> (iii) the final policy with scaffolding $\pi_{\text{RuscaRL}}(\cdot \mid q, R_S)$.
> For each prompt $q$, we compute the rubric score of the response generated by $\pi_{\text{RuscaRL}}(\cdot \mid q)$ and retain only those prompts for which this score exceeds a threshold $\tau$. On this filtered subset, we embed the responses produced by $\pi_{\text{init}}(\cdot \mid q)$, $\pi_{\text{RuscaRL}}(\cdot \mid q)$, and $\pi_{\text{RuscaRL}}(\cdot \mid q, R_S)$, and compute the pairwise cosine distances.
>
> As summarized in Table R3, the distance between $\pi_{\text{init}}(\cdot \mid q)$ and $\pi_{\text{RuscaRL}}(\cdot \mid q)$ remains large (0.30–0.37), while the distance between $\pi_{\text{RuscaRL}}(\cdot \mid q)$ and $\pi_{\text{RuscaRL}}(\cdot \mid q, R_S)$ is substantially smaller (0.12–0.16) and decreases at higher score thresholds. This indicates that, for high-quality outputs, the reasoning patterns encouraged by scaffolding have been largely internalized by the final policy.
> *These additional results have been updated in Appendix C.7 (Page 24) of the revised manuscript.*
>
> Table R3. Semantic distances between responses generated by the initial policy and the final policy, with and without rubric-based scaffolding.
>
> |Threshold $\tau$|dist($\pi_{\text{init}}(\cdot \mid q)$, $\pi_{\text{RuscaRL}}(\cdot \mid q)$)|dist($\pi_{\text{RuscaRL}}(\cdot \mid q)$, $\pi_{\text{RuscaRL}}(\cdot \mid q, R_S)$)|
> |-|-|-|
> |0.5|0.37|0.16|
> |0.8|0.30|0.14|
> |0.9|0.32|0.12|

---

> ### Author Response · Authors · 2025-11-21
> **Response (4/4)**
>
> **[Q4]: Impact of Maximal Scaffolding on Novelty**
>
> Thanks for the suggestion.
> We have additionally conducted experiments to assess whether maximal scaffolding suppresses highly novel solutions. Using a held-out set of 500 instructions, we run the policy separately for each intra-group scaffolding ratio $\lambda_i \in \{1.0, 0.8, 0.6, 0.4, 0.2, 0.0\}$, generating one response per instruction per $\lambda_i$. For each generated sequence, we compute the sequence-level importance ratio $\rho_{\text{seq}}$ and, for each $\lambda_i$, count how many sequences satisfy $\rho_{\text{seq}} > 2$, $\rho_{\text{seq}} > 10$, and $\rho_{\text{seq}} > 100$. We also report, among all sequences with $\rho_{\text{seq}} > 100$, the proportion contributed by each $\lambda_i$.
>
> As shown in Table R4, sequences with extremely high importance ratios ($\rho_{\text{seq}} > 100$) occur at $\lambda_i = 1$, 0.8, 0.6, and 0.4, with the $\lambda_i = 1$ setting contributing 21.0%. In contrast, zero scaffolding ($\lambda_i = 0$) yields only 5.9%. This indicates that both high and moderate scaffolding ratios can produce sequences with very large importance ratios, and maximal scaffolding does not inhibit such behavior.
> *These additional results have been updated in Appendix C.8 (Pages 24–25) of the revised manuscript.*
>
> Table R4. Novelty statistics under different scaffolding ratios.
>
> |$\lambda_i$|$\rho_{\text{seq}} > 2$|$\rho_{\text{seq}} > 10$|$\rho_{\text{seq}} > 100$|Share among $\rho_{\text{seq}} > 100$|
> |-|-|-|-|-|
> |1.0|451|40|25|21.0%|
> |0.8|460|44|24|20.2%|
> |0.6|428|39|21|17.6%|
> |0.4|443|45|24|20.2%|
> |0.2|389|30|18|15.1%|
> |0.0|321|11|7|5.9%|
>
> ---
>
> **References**
>
> [1] HealthBench: Evaluating Large Language Models Towards Improved Human Health. arXiv 2025.
>
> [2] Kimi K2: Open Agentic Intelligence. arXiv 2025.
>
> [3] Baichuan-M2: Scaling Medical Capability with Large Verifier System. arXiv 2025.
>
> [4] DR Tulu: An open, end-to-end training recipe for long-form deep research. https://allenai.org/blog/dr-tulu 2025.
>
> [5] Rubrics as Rewards: Reinforcement Learning Beyond Verifiable Domains. arXiv 2025.
>
> [6] Reinforcement Learning with Rubric Anchors. arXiv 2025.
>
> [7] ResearchQA: Evaluating Scholarly Question Answering at Scale Across 75 Fields with Survey-Mined Questions and Rubrics. arXiv 2025.
>
> [8] ProfBench: Multi-Domain Rubrics Requiring Professional Knowledge to Answer and Judge. arXiv 2025.
>
> [9] ResearchRubrics: A Benchmark of Prompts and Rubrics For Evaluating Deep Research Agents. arXiv 2025.
>
> [10] Rubric-Based Benchmarking and Reinforcement Learning for Advancing LLM Instruction Following. arXiv 2025.

---

> ### Author Response · Authors · 2025-11-27
> **Looking Forward to your Reevaluation**
>
> Dear Reviewer nBsg,
>
> We are glad that the reviewer appreciates our attempt, and sincerely thank you for the constructive comments. As suggested, we have additionally provided detailed clarifications for the focus of RuscaRL and its hyperparameter sensitivity, and included further discussions and experiments on scaffolding internalization and robustness. Please let us know if you have other questions or comments.
>
> Since the discussion window has less than a week remaining, we sincerely look forward to your reevaluation of our work. Thank you very much!
>
> Best regards,
>
> Authors of RuscaRL

---

### Official Review · Reviewer_u4MF · 2025-10-28

**Soundness:** 3
**Presentation:** 3
**Contribution:** 3
**Rating:** 6
**Confidence:** 4

**Summary:**

The paper proposes Rubric-Scaffolded Reinforcement Learning (RuscaRL), aiming to address the exploration bottleneck in open-ended domains for RLVR. It integrates rubric-based external guidance directly into task instructions, creating a controllable exploration schedule. The method is evaluated across multiple open-ended benchmarks (e.g., HealthBench, writing, instruction-following) and multiple models, showing consistent performance gains.

**Strengths:**

1. Motivation is clear and reasonable: the paper directly targets the exploration bottleneck of RLVR in open-ended domains by integrating external guidance into task instructions to improve rollout diversity and quality, enabling controllable, scheduled exploration via rubric scaffolds.

2. Strong empirical evidence: across multiple models and diverse tasks, the method consistently outperforms strong baselines, including Rubric-based RL.

3. Insightful ablations on scaffolding: the ablation studies systematically examine when and how to apply scaffolds, offering practical guidance for maximizing exploration benefits.

**Weaknesses:**

1. Baseline collapse issue (Figure 5): The Rubric-based RL baseline appears to collapse around 200 training steps, with entropy exploding, which raises concerns about the validity of the comparison. It is unclear whether this collapse is caused by non-robust experimental settings or run-specific instability, or whether it reflects an intrinsic weakness of the baseline. Since RuscaRL’s main gains only emerge after 200 steps (with no obvious improvements before that point in Fig. 5b), a substantial portion of the reported improvement might be due to baseline failure rather than method superiority.

    Given that Rubric-based RL is the most crucial baseline, there should be at least one comparison run where the baseline does not collapse to demonstrate that the improvement is robust to make the main claim convincing. Alternatively, the authors should provide a thorough explanation for the baseline collapse.

2. Novelty concerns: The idea of adding external guidance or scaffolding in RL is not entirely new—e.g., similar concepts appear in prior work such as [1]. While RuscaRL extends this to open-ended domains with rubric scheduling, the conceptual contribution is incremental rather than groundbreaking.

[1] MeRF: Motivation-enhanced Reinforcement Finetuning
 for Large Reasoning Models

**Questions:**

Q1. What caused Rubric-based RL’s entropy explosion at step 200? Did this issue occur across all models and seeds, or was it run-specific? Did the authors try tuning or stabilizing the baseline (e.g., different KL, learning rate, sampling parameters/top-p)? Finally, do the reported gains still hold when the baseline remains stable?

---

> ### Author Response · Authors · 2025-11-21
> **Response**
>
> We sincerely appreciate the reviewer for highlighting the strengths of our work, including the clear and reasonable motivation, the strong empirical evidence across models and tasks, and the insightful scaffolding ablations.
> We have carefully revised the manuscript according to your valuable suggestions.
> Below we address the main points raised in the review.
>
> **[W1 & Q1]: Baseline collapse issue**
>
> Thanks for the suggestion.
> **(1) Root cause analysis and multiple runs.**
> We have additionally conducted experiments to diagnose the entropy explosion in Rubric-based RL. As reported in Appendix E.2, repeated runs with different seeds confirm that the collapse consistently appears around 200 steps. The root cause is traced to the sampling parameters in the original setup (temperature = 0.7, top-p = 0.8, top-k = 20). Due to this exploration-limited sampling configuration, the policy entropy drops too quickly, and once it decreases to a threshold (≈0.2), it consistently begins to explode, leading to a collapse in model performance.
> *These additional results have been updated in Appendix E.2 (Page 30) of the revised manuscript.*
>
> **(2) Stabilized baseline with high-temperature sampling.**
> We have additionally conducted experiments to stabilize Rubric-based RL following the reviewer’s suggestion. As detailed in Appendix F, we increased the temperature to 1.0 and removed truncation (top-p = 1.0, top-k = −1). This more exploratory configuration eliminates the sharp entropy explosion and yields smooth, stable training curves across seeds.
>
> **(3) Do the reported gains still hold when the baseline remains stable?**
> Yes. Under the stabilized configuration, Rubric-based RL improves substantially, but RuscaRL still consistently outperforms it throughout training. On HealthBench-500, the final scores are 56.4 (RuscaRL) vs. 52.0 (Rubric-based RL). *These results confirm that the main gains are not due to baseline failure but to the benefits of rubric-scaffolded exploration.*
> *These additional results have been updated in Appendix F (Page 31) of the revised manuscript.*
>
>
>
> **[W2]: Novelty concerns**
>
> Sorry for the confusion.
> While both MeRF and our method augment RLVR with external textual signals, they differ in several key aspects.
> (i) MeRF mainly provides coarse directional "motivation" by injecting the reward description into the prompt [1], whereas RuscaRL uses checklist-style rubrics as structured intermediate scaffolding that decomposes the task into fine-grained criteria and explicitly guides the reasoning process during rollouts.
> (ii) Beyond adding guidance, RuscaRL introduces intra-group scaffolding differentiation and inter-step scaffolding decay, which together promote diverse trajectories within each GRPO group and gradually remove scaffolding so that the model internalizes the reasoning patterns instead of remaining dependent on external guidance.
> *These clarifications have been updated in Related Works (Page 3) of the revised manuscript.*
>
> ---
>
> **References**
>
> [1] MeRF: Motivation-Enhanced Reinforcement Finetuning for Large Reasoning Models. arXiv 2025.

---

> > ### Comment · Reviewer_u4MF · 2025-11-25
> >
> > Thank you for the additional experiments and clarifications. Unfortunately, they do not fully address my concern and in fact raise my concern about the robustness of the results. In Table 2, under the original setting, Rubric-based RL vs. RuscaRL on HealthBench-500 is about 41.2 vs. 50.3 (please correct me if I’m wrong). Under the stabilized setting, the numbers become 52.0 vs. 56.4, so the gain shrinks substantially. The new experiment is only shown for HealthBench-500, which is also your strongest benchmark, while on 6 out of 8 other benchmarks the reported gains over Rubric-based RL are already relatively small (around 0–3 points). It is unclear whether similar stabilization would further reduce those gains, raising the question of whether RuscaRL is robustly beneficial across tasks or mainly effective on a limited subset of benchmarks.
> >
> > This also affects how I interpret your ablations. In Fig. 6(c/d), RuscaRL’s performance appears quite sensitive to the sigmoid decay hyperparameters α and t₀, with only a narrow region around your chosen values achieving the best results. Under the more stable setting where Rubric-based RL itself reaches 52.0 on HealthBench-500, it seems plausible that many of these RuscaRL variants would no longer outperform a strong baseline. This makes the method’s advantage look highly dependent on specific hyperparameter choices.

---

> ### Author Response · Authors · 2025-11-27
> **Response**
>
> Sorry for the confusion.
> (1) To address your concern that the gains under the stabilized setting raise the question of whether RuscaRL is robustly beneficial across tasks or mainly effective on a limited subset of benchmarks, we ran both RuscaRL and Rubric-based RL under the stabilized sampling setting across multiple benchmarks.
> As shown in Table R1, RuscaRL consistently outperforms Rubric-based RL on HealthBench-500, LLMEval-Med, WritingBench, and Creative Writing.
> For the other benchmarks, since they are closed-ended problems with clear ground-truth answers while our training data is entirely open-ended, there exists a substantial train–test bias between the training data and these evaluation benchmarks; therefore, the improvements are less pronounced.
>
> (2) Moreover, to address the concern that RuscaRL’s advantage might be highly sensitive to the sigmoid decay hyperparameters, we also ran additional ablations under the stabilized baseline setting (**see Fig. 11 in the revision**). We would like to emphasize two points:
> - (i) **RuscaRL outperforms the baseline over a wide range of hyperparameters.** Varying $\alpha$ and $t_{0}$ across a broad range yields wide plateaus where RuscaRL remains significantly/clearly above the Rubric-based RL baseline (52.0); only when $\alpha$ and $t_{0}$ are excessively small or large do we observe a pronounced degradation in performance.
> - (ii) **Practical tuning heuristics are simple.** Our experiments suggest the following easy-to-apply guidelines for new tasks or models: Choose $\alpha$ in a broad range around 100 to obtain a reasonably steep but not abrupt decay, and choose $t_{0}$ in [0.15, 0.3], so that strong scaffolding dominates the early 15–30% of training and then starts to decay.
>
> In fact, we used the same set of hyperparameters across all tasks and still achieved strong results, without performing task-specific tuning, which further demonstrates that our method is robust rather than overly sensitive to hyperparameter choices.
> *These additional results have been updated in Appendix F (Page 31) of the revised manuscript.*
>
> (3) Prior work also suggests that RL outcomes can be highly sensitive to rollout decoding temperature [2,3,4]. For example, in Section 5.1 of [2] (Fig. 4), at around 600 training steps on AIME2024, T=1.2 achieves Pass@1≈0.32 while T=0.6 is ≈0.14, a gap of ≈0.18 under otherwise identical settings, highlighting: "Lower temperatures may prematurely limit exploration, especially early in training, while properly higher temperatures enable broader behavioral exploration and improved reward optimization."
> Importantly, the change to optimized sampling parameters benefits both methods. Since the scaffolding mechanism in RuscaRL mainly addresses an exploration deficiency that overlaps with what higher-temperature sampling already mitigates, using more exploratory sampling parameters naturally makes the gain over Rubric-based RL shrink.
>
> (4) We would like to point out that the RuscaRL framework uncovers two key roles of rubrics: **Rubric as Verifiable Rewards for Exploitation** and **Rubric as Explicit Scaffolding for Exploration**. It is worth noting that existing rubric-based RL methods [5,6] are **very recent concurrent work**, all appearing within roughly two months of our ICLR submission deadline. Therefore, we view the strong performance of Rubric-based RL as also aligning with our contribution on **Rubric as Verifiable Rewards**, while RuscaRL further contributes the explicit scaffolding mechanism to improve exploration.
>
> Table R1. Results on multiple benchmarks under the stabilized setting.
>
> |Model|HealthBench-500|LLMEval-Med|MedQA|MedMCQA|WritingBench|Creative Writing|IFEVAL|IFBench|
> |-|-|-|-|-|-|-|-|-|
> |Initial|23.6|47.9|61.8|56.3|45.3|37.4|71.0|26.8|
> |Rubric-based RL|52.0|58.4|63.0|56.4|54.9|38.8|75.2|30.0|
> |RuscaRL|56.4|64.4|64.0|56.4|58.9|40.7|75.3|32.3|
>
> ---
>
> **References**
>
> [2] Scaling Up RL: Unlocking Diverse Reasoning in LLMs via Prolonged Training. arXiv 2025.
>
> [3] AceReason-Nemotron 1.1: Advancing Math and Code Reasoning through SFT and RL Synergy. arXiv 2025.
>
> [4] Enhancing Efficiency and Exploration in Reinforcement Learning for LLMs. EMNLP 2025.
>
> [5] Rubrics as Rewards: Reinforcement Learning Beyond Verifiable Domains. arXiv 2025.
>
> [6] Reinforcement Learning with Rubric Anchors. arXiv 2025.

---

> > ### Comment · Reviewer_u4MF · 2025-11-27
> >
> > Thank you for your response. Intuitively, I believe this method could be helpful in certain scenarios. However, the authors do not have sufficient experimental evidence to demonstrate this.
> > First, in your original submission, you used an unstable training setting in your main table as well as in the comparison with the most important baseline, which I find unacceptable. Second, your new results show that on some benchmarks there is no improvement or only very small gains. You therefore need to provide a more detailed analysis of the conditions under which your method is helpful and when it is not, and revise the paper accordingly.
> >
> > The statement “Since the scaffolding mechanism in RuscaRL mainly addresses an exploration deficiency that overlaps with what higher-temperature sampling already mitigates, using more exploratory sampling parameters naturally makes the gain over Rubric-based RL shrink.” is reasonable. However, many ways of encouraging exploration can be achieved at very low cost simply by tuning hyperparameters, and many of these are already commonly adopted as default training tricks. RuscaRL should demonstrate its advantages under such realistic, exploration-friendly experimental settings, rather than relying on a setting that is particularly favorable to the method (i.e., a lack-of-exploration setting).
> > In light of these issues with the paper, I will unfortunately be lowering my score.

---

> ### Author Response · Authors · 2025-11-27
> **Response**
>
> Thanks for your comment. We would like to clarify some misunderstandings during the discussion phase as follows:
>
> > First, in your original submission, you used an unstable training setting in your main table as well as in the comparison with the most important baseline, which I find unacceptable. Second, your new results show that on some benchmarks there is no improvement or only very small gains.
>
> **(1) On the role of rubric-based RL in our contribution.** The main contribution of RuscaRL is an RL framework that integrates rubrics as (A) explicit scaffolding for exploration and as (B) verifiable rewards for exploitation. **Both (A) and (B) are new relative to the original RLVR recipe.** Therefore, we would first like to clarify that **even rubric-based RL (i.e., (A) using rubrics only as verifiable rewards) is part of our contribution**: this component enables extending RLVR from closed-ended to open-ended domains. Existing rubric-based RL methods [5,6] are **very recent concurrent work**, appearing within roughly 2 months of our ICLR submission deadline. In this way, we should treat rubric-based RL as an **ablation** baseline within our framework, rather than as an external existing method that stands apart from our contribution. Thus, the performance gains from (A) and (B) over the original model are **both** integral parts of our contribution.
>
> **(2) On the impact of sampling parameters on our results.** Our original experiments were designed to **ensure fair and consistent comparisons** across all methods and tasks. To that end, we directly follow a fixed sampling configuration (temperature = 0.7, top-p = 0.8, and top-k = 20) **recommended by the Qwen team** [7]. This configuration was applied uniformly across all models and benchmarks without any method-specific or task-specific tuning. We therefore believe that comparisons under this unified setting are both valid and meaningful.
> - We appreciate the reviewer's suggestion, which motivated us to explore better sampling hyperparameters. We are glad to see that with additional hyperparameter tuning, **all methods improve**, and RuscaRL continues to provide promising gains, especially on the open-ended benchmarks that are the main focus of our work.
> For example, in Table R1, RuscaRL (A and B) and Rubric-based RL (Only A ablation) vs original achieve *56.4 and 52.0 v.s. 23.6* on HealthBench-500, *64.4 and 58.4 v.s. 47.9* on LLMEval-Med, *58.9 and 54.9 v.s. 45.3* on WritingBench, *40.7 and 38.8 v.s. 37.4* on Creative Writing, *75.3 and 75.2 v.s. 71.0* on IFEVAL, and *32.3 and 30.0 v.s. 26.8* on IFBench.
> - We would also like to clarify that our paper primarily **targets the open-ended domain**: our training data consists entirely of open-ended tasks, whereas MedQA and MedMCQA are **closed-ended, multiple-choice benchmarks**. Our performance on these closed-ended benchmarks in Table R1 show only marginal improvements over the base model, and we do not claim large gains there. These results are included mainly to demonstrate cross-task generalization, rather than as the main evidence for our contribution. We will revise the paper to make this focus and the associated limitations on closed-ended benchmarks more explicit (Section 5.2).
>
> > Many ways of encouraging exploration can be achieved at very low cost simply by tuning hyperparameters, and many of these are already commonly adopted as default training tricks.
>
> **(3) On the relationship between hyperparameter tuning and our scaffolding mechanism.** Tuning sampling hyperparameters is a simple and low-cost way to encourage exploration. However, as noted in our previous response, prior work also reports that model performance can be quite sensitive to these hyperparameters [2,3,4,8]. Thus, relying purely on hyperparameter tuning can be brittle and may not provide a robust, general solution. In this work, hyperparameter tuning and our scaffolding mechanism are **orthogonal**. These two mechanisms act differently: sampling hyperparameters mainly increase **undirected stochasticity**, whereas our scaffolding mechanism introduces **rubric-conditioned, structured exploration** that steers learning toward high-quality trajectories. Empirically, as shown in both our original experiments and the new results under different sampling configurations, RuscaRL consistently yields additional gains, indicating that scaffolding is complementary to, rather than a substitute for, hyperparameter tuning.
>
> ---
>
> References
>
> [7] Qwen3 Technical Report. arXiv 2025.
>
> [8] Understanding Hyperparameter Effects in Open-Ended Text Generation. COLING 2025.

---

> ### Author Response · Authors · 2025-11-27
> **Thanks for the Comments**
>
> We thank the reviewer for the continued interest in RuscaRL and for the insightful comments. Regarding your concerns about the experimental settings, we respectfully hope that this clarification addresses your doubts. We are more than willing to revise the manuscript to ensure this point is crystal clear for future readers.
> If these updates lead you to see the work differently, we would be very thankful if you could reconsider the evaluation, and we truly appreciate the time and care you have devoted to our submission.

---

### Official Review · Reviewer_FzkW · 2025-10-29

**Soundness:** 3
**Presentation:** 3
**Contribution:** 2
**Rating:** 4
**Confidence:** 4

**Summary:**

The paper proposes RuscaRL, a rubric-scaffolded reinforcement learning framework for LLMs. It introduces (1) intra-group rubric differentiation to encourage diverse rollouts and (2) a sigmoid inter-step schedule to gradually remove rubric hints. Rubric criteria are reused as binary “verifiable” rewards via another LLM judge. The method is evaluated on medical QA, instruction following, writing, and STEM tasks.

**Strengths:**

1. The empirical results show improvements across several domains and model sizes.

2. The motivation (mitigating exploration bottlenecks in RL for LLMs) is interesting and relevant.

3. Different ablation studies are included.

4. The results are overall interesting.

**Weaknesses:**

1. Overstated novelty: Rubric rewards, scaffolded prompting, and gradual hint removal resemble prior work in rubric-guided RL, chain-of-thought prompting, and curriculum learning. The related work section cites almost exclusively 2025 works; the absence of foundational literature (curriculum RL, reward shaping, structured exploration strategies) is problematic.

2. Related-work coverage: Classical literature on exploration, curriculum RL, and scaffolding is missing.

3. Insufficient exploration baselines: No comparison against established exploration strategies (e.g., entropy-guided branching, curriculum RL variants, prolonged compute), making it hard to support claims about breaking exploration bottlenecks.

4. Lack of statistical rigor: Results are reported without standard deviations or confidence intervals, despite typical high variance in RL settings.

5. Dependence on rubric quality: The method assumes rubrics are complete, well-weighted, and unbiased. No analysis of noisy, incomplete, or contradictory criteria is provided.

6. LLM-as-judge fragility: Evaluation relies on a single judge model without cross-judge agreement, calibration, or human validation, risking optimization toward judge quirks.

7. Shallow compliance risk: Binary criteria may reward formatting or keyword heuristics rather than genuine reasoning, but no analysis investigates this failure mode.

8. Figure and clarity issues: Figure 1 (left) is not informative, and Figure 2 is understandable only after reading the text carefully; simpler visualizations could improve clarity.

**Questions:**

1.	How does performance change when swapping the judge model? Do results remain stable?

2.	What happens if rubric criteria are reordered or partially corrupted?

3.	Can the authors compare against entropy-based approaches or curriculum RL as well as recent exploration approaches under matched compute?

4.	Can you provide standard deviations or confidence intervals for all tables?

---

> ### Author Response · Authors · 2025-11-21
> **Response (1/6)**
>
> We sincerely appreciate the reviewer for highlighting the strengths of RuscaRL, including the improvements across domains and model sizes, and the interesting motivation and results.
> We have carefully revised the manuscript according to your valuable suggestions.
> Below we address the main points raised in the review.
>
>
> **[W1 & W2]: Overstated novelty and missing classical literature**
>
>
> Thanks for your valuable comment. We would like to clarify how RuscaRL differs from these related directions as follows:
>
> (1) Compared with rubric-guided RL. It is worth noting that existing rubric-guided RL methods [1,2] are **very recent concurrent work**, all appearing within roughly two months of our ICLR submission deadline. These rubric-guided RL methods **only use rubrics as verifiable rewards**, and RL is then applied on top of standard rollout generation. In contrast, RuscaRL **introduces rubrics as explicit scaffolding** during rollout generation rather than merely specifying a reward for optimization. This design improves both the diversity and the quality of rollouts, thereby enabling more efficient exploration during RL.
>
> (2) Compared with CoT prompting. CoT prompting often uses a **fixed prompt** (e.g., "think step by step") with the sole purpose of eliciting step by step reasoning [3,4,5,6,7]. In contrast, RuscaRL introduces checklist-style rubrics as prompts with two advantages:
> (a) they provide **multi-dimensional criteria** such as factuality, structure, completeness, and safety, enabling exploration along multiple aspects rather than a single step-wise format; and
> (b) they support a **dynamic schedule** where different subsets of criteria are assigned across samples and gradually reduced over training, allowing the model to internalize these patterns without relying on persistent prompts.
>
> (3) Compared with curriculum learning. Curriculum learning often relies on **predefined/predicted task difficulty**, presenting simpler tasks earlier and harder tasks later [8,9,10,11,12]. In contrast, RuscaRL does not require estimating or ordering tasks by difficulty. We **keep the task distribution unchanged and introduce a straightforward curriculum over rubric visibility**: different subsets of checklist criteria are assigned to different samples within each GRPO group, and the overall amount of scaffolding decreases over training. This rubric-based curriculum enables efficient exploration without assuming any prior difficulty structure, and the decay schedule allows the policy to gradually remove its dependence on rubric guidance.
>
> *We have updated Related Works (Page 3) of the revised manuscript to include the missing classical literature.*
>
>
> **[W3 & Q3]: Insufficient exploration baselines**
>
> Thanks for the suggestion.
> We have additionally conducted experiments that incorporate exploration-oriented baselines for RL with LLMs under matched compute and data. Specifically, we compare against:
> (i) RL-Plus [13], a hybrid policy-optimization method designed to mitigate capability boundary collapse;
> (ii) Entropy-based RL [14], which modifies the advantage by adding a clipped, gradient-detached token-level entropy bonus to induce exploration through higher-entropy reasoning trajectories;
> (iii) Curriculum RL [8,9,10,11,12], where we precompute rubric scores from a single pass of the base model, sort examples from easy to hard, and train with this fixed curriculum by disabling data shuffling;
> (iv) ProRL [15], for which we follow the prolonged-training regime using DAPO [16], running 1,000 RL steps with rollout temperature 1.2 to induce stronger exploration.
>
> We train all exploration-oriented baselines and RuscaRL on medical-domain data using Qwen2.5-7B-Instruct, and evaluate them on four medical benchmarks.
> As shown in Table R1, RuscaRL achieves the best performance among all methods, delivering consistent gains across the four medical benchmarks under matched compute and data.
> *These additional results have been updated in Appendix C.9 (Page 25) of the revised manuscript.*
>
> Table R1. Comparison of exploration-oriented RL baselines on medical benchmarks.
>
> |Method|HealthBench-500|LLMEval-Med|MedQA|MedMCQA|
> |-|-|-|-|-|
> |Initial Model|23.4±0.3|48.0±0.3|61.8±0.2|56.3±0.1|
> |Rubric-based RL|41.1±0.1|54.6±0.2|62.1±0.4|56.3±0.1|
> |RL-Plus|45.1±0.5|58.4±0.2|62.0±0.1|56.3±0.1|
> |Entropy-based RL|42.2±0.2|57.0±0.7|62.8±0.1|56.6±0.2|
> |Curriculum RL|40.3±0.4|56.1±0.5|62.4±0.2|56.4±0.1|
> |ProRL|49.9±0.3|60.0±0.3|62.1±0.4|56.2±0.3|
> |**RuscaRL**|**50.3±0.4**|**61.2±0.5**|**63.5±0.1**|**56.5±0.1**|

---

> ### Author Response · Authors · 2025-11-21
> **Response (2/6)**
>
> **[W4 & Q4]: Lack of statistical rigor**
>
>
> Sorry for the confusion.
> **(1) Evaluation variability (standard deviations).**
> We have additionally conducted three independent evaluation runs for each result and report the corresponding standard deviations in Tables R2 and R3.
> Since HealthBench-500, LLMEval-Med, and WritingBench were each evaluated only once in the initial submission due to cost considerations (see Appendix B.2 for detailed evaluation settings), their current mean scores show slight differences compared with those previously reported in the main experiments.
> *These additional results have been updated in Appendix E.1 (Pages 29-30) of the revised manuscript.*
>
> **(2) Training stability across random seeds.**
> We have additionally conducted experiments in which we repeat the training of RuscaRL and rubric-based RL with five random seeds under the same experimental setup as in the main experiments.
> The resulting training curves (Appendix E, Fig. 9) show high consistency across seeds, confirming that RuscaRL’s improvements are not attributable to random chance.
> *These additional results have been updated in Appendix E.2 (Page 30) of the revised manuscript.*
>
> Table R2. Extended version of Table 1 in the main paper, with empirical standard deviations added.
>
> |Model|HealthBench-500|LLMEval-Med|MedQA|MedMCQA|WritingBench|Creative Writing|IFEVAL|IFBench|
> |-|-|-|-|-|-|-|-|-|
> |Qwen3-30B-A3B-Instruct|46.9±0.3|71.5±0.3|84.2±0.2|71.3±0.1|78.1±0.3|74.4±0.5|83.0±0.4|31.9±0.5|
> |+ RuscaRL|61.1±0.2|73.2±0.4|84.8±0.3|71.9±0.2|79.2±0.1|74.3±0.3|84.5±0.1|32.1±0.0|
> |Qwen3-30B-A3B-Base|11.2±0.5|43.1±0.6|73.6±0.1|65.1±0.4|36.9±1.2|35.8±2.0|39.0±0.7|13.3±0.5|
> |+ RuscaRL|48.4±0.4|60.9±0.2|71.3±0.4|65.4±0.2|59.5±1.0|46.0±1.0|76.3±0.5|30.3±0.7|
> |Qwen2.5-7B-Instruct|23.4±0.3|48.0±0.3|61.8±0.2|56.3±0.1|45.2±0.9|37.4±0.9|71.0±0.5|26.8±0.3|
> |+ RuscaRL|50.3±0.4|61.2±0.5|63.5±0.1|56.5±0.1|56.1±0.3|38.6±0.6|75.3±0.0|31.0±0.3|
> |Qwen2.5-7B|8.5±1.2|28.2±0.4|55.3±0.1|55.0±0.2|23.8±0.9|30.3±1.6|32.0±0.3|14.5±0.4|
> |+ RuscaRL|46.3±0.4|47.9±0.2|58.2±0.4|55.6±0.4|46.0±1.1|34.8±1.0|56.2±0.3|25.9±0.2|
> |Llama-3.1-8B-Instruct|12.5±0.8|30.1±0.5|66.8±9.1|58.0±0.2|36.7±0.4|44.5±0.2|72.6±0.6|22.6±0.6|
> |+ RuscaRL|46.0±0.2|46.2±0.5|70.7±0.2|60.7±0.1|52.7±0.1|54.2±0.7|79.7±0.0|31.1±0.1|
> |Llama-3.1-8B|0.0±0.0|9.1±0.3|36.9±0.3|35.9±0.2|13.0±0.7|26.3±1.9|18.1±1.0|11.6±1.2|
> |+ RuscaRL|25.8±0.2|29.6±0.3|49.7±0.3|45.4±0.2|35.7±0.3|33.3±1.0|55.6±1.0|21.4±1.1|
>
> Table R3. Extended version of Table 2 in the main paper, with empirical standard deviations over three evaluation runs added.
>
> |Model / Method|HealthBench-500|LLMEval-Med|MedQA|MedMCQA|WritingBench|Creative Writing|IFEVAL|IFBench|
> |-|-|-|-|-|-|-|-|-|
> |Qwen2.5-7B-Instruct|23.4±0.3|48.0±0.3|61.8±0.2|56.3±0.1|45.2±0.9|37.4±0.9|71.0±0.5|26.8±0.3|
> |├─ Rubric-based RL|41.1±0.1|54.6±0.2|62.1±0.4|56.3±0.1|53.7±0.4|38.8±1.0|75.1±0.4|29.3±0.4|
> |├─ Rubric-based RL-S|36.8±0.6|56.1±0.7|57.9±0.3|52.4±0.4|45.9±0.2|38.3±1.1|71.9±0.5|28.6±0.4|
> |├─ RuscaRL (Ours)|50.3±0.4|61.2±0.5|63.5±0.1|56.5±0.1|56.1±0.3|38.6±0.6|75.3±0.0|31.0±0.3|
> |├─ SFT|38.3±0.2|52.6±0.2|60.8±0.1|57.3±0.4|62.8±0.2|45.3±0.6|75.2±0.1|25.2±0.6|
> |├─ SFT + Rubric RL|55.5±0.5|58.5±0.1|59.7±0.2|56.4±0.2|66.7±0.1|43.6±0.7|82.1±0.5|29.6±0.1|
> |└─ SFT + RuscaRL|56.9±0.1|58.8±0.2|61.6±0.1|56.9±0.1|67.0±0.5|43.9±0.6|82.5±0.3|30.6±0.5|
> |Qwen2.5-7B|8.5±1.2|28.2±0.4|55.3±0.1|55.0±0.2|23.8±0.9|30.3±1.6|32.0±0.3|14.5±0.4|
> |├─ Rubric-based RL|42.0±0.5|46.5±0.2|48.2±0.3|49.9±0.4|40.1±0.5|33.8±1.5|53.4±0.5|25.5±0.7|
> |├─ Rubric-based RL-S|21.7±0.2|44.4±0.6|60.3±0.2|55.5±0.2|43.4±0.5|25.7±1.9|52.3±0.1|20.4±0.8|
> |├─ RuscaRL (Ours)|46.3±0.4|47.9±0.2|58.2±0.4|55.6±0.4|46.0±1.1|34.8±1.0|56.2±0.3|25.9±0.2|
> |├─ SFT|32.2±0.2|40.0±0.1|56.5±0.1|54.4±0.0|56.6±0.1|42.5±0.8|69.7±0.4|20.8±0.3|
> |├─ SFT + Rubric RL|36.5±0.4|39.7±0.5|57.1±0.1|54.1±0.1|57.4±0.4|43.2±0.7|71.6±0.5|23.7±0.4|
> |└─ SFT + RuscaRL|35.4±0.1|42.7±0.1|58.2±0.2|55.1±0.0|57.7±0.3|42.6±0.8|72.0±0.1|23.1±0.1|

---

> ### Author Response · Authors · 2025-11-21
> **Response (3/6)**
>
> **[W5 & Q2]: Dependence on rubric quality**
>
> Thanks for the valuable comment.
> We have additionally conducted experiments to inform practical application, perturbing the rubrics to evaluate how robust our method is under noisy, incomplete, or even contradictory criteria.
> For each rubric, we design the following noise variants:
> - **Original**: the unmodified rubric.
> - **Inverse**: swap high-point and low-point criteria, effectively reversing the relative scoring priorities.
> - **Negated**: flip the sign of each criterion score (e.g., +3 → −3), so "good" behavior is penalized and "bad" behavior is rewarded.
> - **Ambiguous**: inject vague or subjective criteria generated by GPT-4.1.
> - **Contradictory**: inject logically conflicting criteria generated by GPT-4.1.
> - **50% removed**: randomly delete 50% of the original criteria, simulating rubrics with substantially incomplete coverage.
>
> For each noise setting, we train a Rubric-based RL baseline and RuscaRL on medical-domain data using Qwen2.5-7B-Instruct, and evaluate them on four medical benchmarks.
> Table R4 shows that RuscaRL is more robust to rubric noise: under mild perturbations (Ambiguous, Contradictory, 50% removed), it consistently outperforms Rubric-based RL, whereas under severe corruptions (Inverse, Negated) both methods degrade substantially.
> *These additional results have been updated in Appendix C.10 (Pages 25-26) of the revised manuscript.*
>
> Table R4. Robustness to rubric noise on medical benchmarks.
> |Rubric variant|HealthBench-500|LLMEval-Med|MedQA|MedMCQA|
> |-|-|-|-|-|
> |Initial Model|23.4±0.3|48.0±0.3|61.8±0.2|56.3±0.1|
> |+Rubric-based RL|||||
> |├─ Original|41.1±0.1|54.6±0.2|62.1±0.4|56.3±0.1|
> |├─ Inverse|7.1±0.2|41.3±0.5|61.3±0.1|55.8±0.1|
> |├─ Negated|2.9±0.5|36.8±1.4|60.5±0.0|55.6±0.1|
> |├─ Ambiguous|40.1±1.0|54.8±1.2|63.0±0.1|56.2±0.2|
> |├─ Contradictory|43.4±0.7|55.9±1.1|63.3±0.5|55.9±0.2|
> |└─ 50% removed|39.6±1.5|51.7±0.9|62.5±0.6|56.7±0.1|
> |+**RuscaRL (Ours)**|||||
> |├─ Original|**50.3±0.4**|**61.2±0.5**|**63.5±0.1**|**56.5±0.1**|
> |├─ Inverse|10.6±0.7|44.4±0.5|61.3±0.2|56.1±0.1|
> |├─ Negated|6.7±0.5|41.1±0.5|60.8±0.0|55.9±0.0|
> |├─ Ambiguous|46.2±0.3|59.6±1.4|62.2±0.2|56.0±0.1|
> |├─ Contradictory|45.7±0.7|56.8±0.6|63.2±0.5|56.2±0.1|
> |└─ 50% removed|44.6±0.2|54.9±0.6|63.2±0.2|56.3±0.1|

---

> ### Author Response · Authors · 2025-11-21
> **Response (4/6)**
>
> **[W6 & Q1]: LLM-as-a-judge fragility and swapping the judge model**
>
> Sorry for the confusion.
> (1) We follow prior work and evaluate on rubric-based benchmarks whose pipelines have already been carefully validated via cross-judge validation, calibration, and human studies [17,18].
> In particular, HealthBench [17] uses multi-stage expert review and calibration and reports that GPT-4.1 as a rubric-based judge reaches macro F1 of about 0.71 versus clinicians, comparable to clinician–clinician consistency (see Table R5 for a comparison across grader models reproduced from [17]).
> While WritingBench [18] systematically studies evaluation schemes and shows that dynamic, query-dependent rubrics with a specialized critic achieve about 83% consistency with human preferences, substantially higher than static rubric baselines.
> Independent rubric-based benchmarks such as ResearchQA [19], ProfBench [20], and ResearchRubrics [21] further document strong LLM human alignment.
> To further verify that different graders are similarly aligned with human annotations, we also computed macro F1 on a subset of 100 examples from our training data and found consistently high agreement across a range of judge models (see Table R6).
>
> Table R5. Agreement (macro F1) between rubric-based LLM judges and clinicians on HealthBench, reproduced from [17].
>
> |Grader model|Macro F1 vs clinicians|
> |-|-|
> |GPT-4.1|0.709|
> |o4-mini|0.692|
> |o3|0.681|
> |GPT-4.1 mini|0.661|
> |GPT-4.1 nano|0.580|
>
> Table R6. Agreement (macro F1) between different grader models and human labels on a subset of 100 examples from our training data.
>
> |Grader model|Macro F1 vs human labels|
> |-|-|
> |GPT-4.1|0.688|
> |Qwen3-32B (thinking)|0.654|
> |Qwen3-32B (non-thinking)|0.644|
> |Qwen3-30B-A3B-Instruct-2507|0.634|
> |Qwen3-4B-Instruct-2507|0.615|
>
> (2) We have additionally conducted experiments that change the grader from Qwen3-32B (non-thinking) to a more efficient Qwen3-30B-A3B-Instruct-2507. The HealthBench-500 score changes only slightly (50.3 vs. 48.9; see Table R7), while at the same time we significantly reduce the extra judge-side computation cost during training, indicating that **RuscaRL’s gains are not tied to a specific grader**. Moreover, a key property of our rubrics is that their criteria are concise and easy to check, so the judgments do not heavily rely on the raw capability of the judge model—moderately strong models already provide reliable rubric-based rewards.
> *These additional results have been updated in Appendix C.6 (Pages 23-24) of the revised manuscript.*
>
> Table R7. Training cost and HealthBench-500 performance with different grader models used as LLM-as-a-judge.
>
> |Grader|Reward Time per training step (s)|Total reward GPU hours|HealthBench-500 score|
> |-|:-:|:-:|:-:|
> |Qwen3-32B (non-thinking)|60|46.7|50.3|
> |Qwen3-30B-A3B-Instruct-2507|18|14.0|48.9|

---

> ### Author Response · Authors · 2025-11-21
> **Response (5/6)**
>
> **[W7]: Shallow compliance risk**
>
>
> Thank you for the insightful comment.
> Conventional rule-based verification indeed can lead to shallow compliance, since programmatic rules often rely on formatting, keyword cues, or other superficial heuristics [22].
> However, our used LLM-as-a-judge rubrics are specifically intended to mitigate this shallow compliance risk.
> This is because
> (i) using an LLM judge with binary, per-criterion decisions leverages a decomposition of the overall score into many simple checks, making each criterion easy to assess in isolation and enabling the LLM to produce more robust and reliable scores;
> (ii) the rubric consists of multiple independent yet complementary criteria, so any shallow shortcut that exploits one criterion is exposed and penalized by others, reducing the overall score;
> (iii) and the rubric provides explicit negative signals against patterned, templated, or unsupported reasoning, directly discouraging shallow compliance.
>
>
> As suggested, we have additionally conducted two analyses in Appendix G that contrast the rule-based shallow-compliance failure mode with the robustness of rubric-based evaluation.
> Appendix G.1 and Appendix G.2 examine two instruction-following tasks where simple rule-based checks are easily hacked through patterned keyword repetition, making the shallow-compliance failure mode explicit.
> Appendix G.3, Appendix G.4 and Appendix G.5 then present a case study assessing whether our approach actually exhibits this issue, showing that the trained model produces genuinely higher-quality responses rather than merely satisfying superficial criteria.
>
> > **Shallow compliance under keyword-based rule (from revised manuscript Appendix G.1)**
> >
> > **Instruction:** “Write a one-sentence restaurant review using the word *delicious* at least five times.”
> >
> > **Rule-hacking answer (passes keyword rule):**
> > The food was delicious, delicious, delicious, delicious, delicious, delicious, delicious, delicious, delicious, delicious, delicious, delicious, delicious, delicious, delicious, delicious, delicious, delicious, delicious, delicious, delicious.
> >
> > **Rubric-aligned answer (genuine satisfaction):**
> > This small restaurant serves delicious soup, delicious noodles, delicious dumplings, delicious desserts, and delicious drinks, and its cozy atmosphere and friendly staff make every delicious meal relaxing and delicious for family dinners.
>
> **[W8]: Figure and clarity issues**
>
> Thanks for the suggestion.
> - For **Figure 1 (left)**, we have added more textual annotations so that the conceptual illustration becomes more informative and easier to understand on its own.
> - For **Figure 2**, we have redesigned the framework into two panels that directly match the two uses of rubrics in RuscaRL ("Rubric as Verifiable Rewards for Exploitation" and "Rubric as Explicit Scaffolding for Exploration") and added clearer labels to improve clarity.
>
> *These figures have been updated in Introduction (Page 2) and Methodology (Page 4) of the revised manuscript.*

---

> ### Author Response · Authors · 2025-11-21
> **Response (6/6)**
>
> **References**
>
> [1] Rubrics as Rewards: Reinforcement Learning Beyond Verifiable Domains. arXiv 2025.
>
> [2] Reinforcement Learning with Rubric Anchors. arXiv 2025.
>
> [3] Chain-of-Thought Prompting Elicits Reasoning in Large Language Models. NeurIPS 2022.
>
> [4] Large Language Models are Zero-Shot Reasoners. NeurIPS 2022.
>
> [5] Least-to-most Prompting Enables Complex Reasoning in Large Language Models. ICLR 2023.
>
> [6] React: Synergizing Reasoning and Acting in Language Models. ICLR 2023.
>
> [7] Tree of Thoughts: Deliberate Problem Solving with Large Language Models. NeurIPS 2023.
>
> [8] Curriculum Learning. ICML 2009.
>
> [9] Curriculum Reinforcement Learning from Easy to Hard Tasks Improves LLM Reasoning. arXiv 2025.
>
> [10] Curriculum Learning: A Regularization Method for Efficient and Stable Billion-Scale GPT Model Pre-Training. arXiv 2021.
>
> [11] Do Data-based Curricula Work? INSIGHTS 2022.
>
> [12] Strategic Data Ordering: Enhancing Large Language Model Performance through Curriculum Learning. arXiv 2024.
>
> [13] RL-Plus: Countering Capability Boundary Collapse of LLMs in Reinforcement Learning with Hybrid-Policy Optimization. arXiv 2025.
>
> [14] Reasoning with Exploration: An Entropy Perspective. arXiv 2025.
>
> [15] ProRL: Prolonged Reinforcement Learning Expands Reasoning Boundaries in Large Language Models. arXiv 2025.
>
> [16] DAPO: An Open-Source LLM Reinforcement Learning System at Scale. arXiv 2025.
>
> [17] HealthBench: Evaluating Large Language Models Towards Improved Human Health. arXiv 2025.
>
> [18] WritingBench: A Comprehensive Benchmark for Generative Writing. arXiv 2025.
>
> [19] ResearchQA: Evaluating Scholarly Question Answering at Scale Across 75 Fields with Survey-Mined Questions and Rubrics. arXiv 2025.
>
> [20] ProfBench: Multi-Domain Rubrics Requiring Professional Knowledge to Answer and Judge. arXiv 2025.
>
> [21] ResearchRubrics: A Benchmark of Prompts and Rubrics For Evaluating Deep Research Agents. arXiv 2025.
>
> [22] Rubric-Based Benchmarking and Reinforcement Learning for Advancing LLM Instruction Following. arXiv 2025.

---

> ### Author Response · Authors · 2025-11-27
> **Looking Forward to your Reevaluation**
>
> Dear Reviewer FzkW,
>
> We are glad that the reviewer appreciates our attempt, and sincerely thank you for the constructive comments. As suggested, we have additionally provided detailed clarifications for LLM-as-a-judge fragility and shallow compliance risk, and included further discussions and experiments on exploration baselines and failure mode. Please let us know if you have other questions or comments.
>
> Since the discussion window has less than a week remaining, we sincerely look forward to your reevaluation of our work and would very appreciate it if you could raise your score to boost our chance of more exposure to the community. Thank you very much!
>
> Best regards,
>
> Authors of RuscaRL

---

### Official Review · Reviewer_CLiv · 2025-11-01

**Soundness:** 2
**Presentation:** 2
**Contribution:** 2
**Rating:** 2
**Confidence:** 3

**Summary:**

This study investigated the RL exploration issue, introducing the Rubric-Scaffolded Reinforcement Learning (RuscaRL) framework designed to overcome the exploration bottleneck in reinforcement learning (RL) for large language models (LLMs). Traditional RL for LLM reasoning struggles because learning high-quality reasoning requires exploration, yet the model’s exploration capacity is limited by its own reasoning ability. Empirically, the approach shows faster reward improvement, higher diversity, and better generalization than baselines.

**Strengths:**

1. The integration of instructional scaffolding method into RL for LLMs is highly good.


2. Demonstrates consistent gains across diverse benchmarks (HealthBench, MedQA, MMLU-Pro, Creative Writing, IFBench, etc.).


3. RuscaRL shows better sample efficiency (steeper Best-of-N curve) and avoids entropy collapse, evidencing better exploration control.

**Weaknesses:**

1. The method’s success hinges on well-designed, domain-specific rubrics. Poorly constructed rubrics may bias training or limit diversity.
2. The paper acknowledges this but does not propose automated rubric construction or robustness checks.


3. Using rubric-guided multi-sample rollouts with LLM-as-a-Judge evaluations is computationally expensive, especially for large-scale training (many rollouts × multiple criteria × grader calls).


4. While results are strong across domains, the framework’s behavior under noisy or conflicting rubrics is not deeply analyzed.

**Questions:**

1. How sensitive is RuscaRL to rubric design noise or inconsistency across domains (e.g., mixing medical and creative writing rubrics)?


2. What are the computational and inference costs compared to standard pipelines?

---

> ### Author Response · Authors · 2025-11-21
> **Response (1/4)**
>
> We sincerely appreciate the reviewer for highlighting the strengths of RuscaRL, including the effective integration of instructional scaffolding, the consistent gains across benchmarks, and the improved sample efficiency and exploration control.
> We have carefully revised the manuscript according to your valuable suggestions.
> Below we address the main points raised in the review.
>
> **[W1 & W2]: Dependence on well-designed rubrics and lack of automated construction or robustness checks**
>
> Sorry for the confusion.
> (1) We would like to clarify that our work does **not** focus on **how to generate rubrics**, but rather on **how to use given rubrics more effectively**. RuscaRL provides a methodological contribution rather than a new dataset. The main contribution of RuscaRL is a reinforcement learning framework that integrates rubrics as explicit scaffolding for exploration and as verifiable rewards for exploitation. Using the given rubric datasets, RuscaRL achieves substantial improvements over other rubric-based RL baselines.
>
> (2) It is also worth noting that rubrics have recently emerged as a promising solution for open-ended domains in the research community [1,2,3,4,5,6]. Unlike mathematics or code with objectively verifiable answers, many real-world tasks such as medical consultation and creative writing often require multidimensional evaluation and lack a single, verifiable ground truth. Although rubrics are often well-designed and domain-specific, **these very properties in fact enable them to provide stable and informative signals** for both training and evaluation.
> We view this as a **worthwhile tradeoff**, especially given their broad potential in open-ended applications. For instance, OpenAI introduced HealthBench [1] to evaluate medical dialogue using rubrics, and leading models such as Kimi K2 and Baichuan M2 integrate rubric-based rewards in reinforcement learning for open-ended tasks [2,3]. These trends highlight that the benefits of rubrics outweigh their limitations.
>
>
> (3) Some recent studies also focus on improving rubric construction itself [5,6,7,8,9,10], and these studies provide valuable advances for building higher-quality evaluation criteria. **Our method is orthogonal to these dataset efforts**, since RuscaRL can integrate with any rubric dataset, regardless of how the rubrics are generated. We hope that our methodological contribution, together with these advances in rubric datasets, will jointly help drive further progress in rubric-based research and promote their effective use in open-ended tasks.

---

> ### Author Response · Authors · 2025-11-21
> **Response (2/4)**
>
> **[W3 & Q2]: Computational cost of rubric-based RL**
>
> Thanks for your insightful comment.
> In our experiments, the policy model (e.g., Qwen2.5-7B-Instruct) is trained on one 8×H200 node, and the Grader model (Qwen3-32B, non-thinking) on an additional node. For each step, we use a batch size of 64 instructions, 8 rollouts per instruction, and an average of 11.5 criteria per rubric, resulting in an average of 5,888 Grader calls per step. The reward stage takes ~60s per step, while the policy computation takes ~40s for rollout and ~15s for update. These costs are comparable to other rubric-based RL baselines using LLMs as judges with multi-criteria scoring (e.g., RaR [5] and Rubicon [6]). As noted in our response to [W1 & W2], the rubrics provide stable and informative signals for training. Although rubric-based rewards introduce roughly a twofold increase in training cost (a limitation shared by all rubric-based methods rather than specific to ours), we believe this cost is well worth it given the strong performance gains on open-ended tasks.
>
> Moreover, our previous implementation was method-focused and not heavily optimized. We have observed in follow-up runs that significant efficiency gains are possible with relatively simple modifications:
>
> **(1) Lightweight grader models.** We have additionally conducted experiments by replacing Qwen3-32B with the lightweight grader Qwen3-30B-A3B-Instruct-2507. The results in Table R1 show that this modification reduces the per-step reward-stage wall-clock time from 60s to 18s, with only a slight degradation in final performance.
>
>
> Table R1. Training cost and HealthBench-500 performance with different grader models.
> |Grader|Reward Time per Training Step (s)|HealthBench-500 Score|
> |-|:-:|:-:|
> |Qwen3-32B (non-thinking)|60|50.3|
> |Qwen3-30B-A3B-Instruct-2507|18|48.9|
>
>
> **(2) Asynchronous rollout-reward strategy.**
> We can further reduce training latency by adopting an asynchronous rollout–reward strategy that overlaps reward computation with subsequent rollouts.
> In the default synchronous pipeline, the per-step latency is:
> $$
> T_{\text{sync}} = T_{\text{rollout}} + T_{\text{reward}} + T_{\text{update}} = 40 + 60 + 15 = 115\text{ s/step}.
> $$
> When the reward stage is run **asynchronously**: each generated sequence is sent to the grader immediately, and grading is overlapped with subsequent rollouts. Then the per-step latency becomes:
> $$
> T_{\text{async}} = \max(T_{\text{rollout}}, T_{\text{reward}}) + T_{\text{update}}.
> $$
>
> Under the same configuration, this reduces the wall-clock time to:
> $$
> T_{\text{async}} = \max(40, 60) + 15 = 75\text{ s/step}.
> $$
>
> With the more efficient Qwen3-30B-A3B grader ($T_{\text{reward}}= 18\text{ s}$), the latency further drops to:
> $$
> T_{\text{async}} = \max(40, 18) + 15 = 55\text{ s/step}.
> $$
> At this point, the latency of the reward stage can be significantly reduced.
>
> *These clarifications have been updated in Appendix C.6 (Pages 23-24) of the revised manuscript.*

---

> ### Author Response · Authors · 2025-11-21
> **Response (3/4)**
>
> **[W4 & Q1]: Behavior under noisy or conflicting rubrics and cross-domain inconsistency**
>
> Thanks for the valuable comment.
> **(1) Robustness to noisy or conflicting rubrics.**
> We have additionally conducted experiments where we perturbed the rubrics to evaluate how robust our method is across different noise conditions. For each rubric, we design the following noise variants:
> - **Original**: the unmodified rubric.
> - **Inverse**: swap high-point and low-point criteria, effectively reversing the relative scoring priorities.
> - **Negated**: flip the sign of each criterion score (e.g., +3 → −3), so "good" behavior is penalized and "bad" behavior is rewarded.
> - **Ambiguous**: inject vague or subjective criteria generated by GPT-4.1.
> - **Contradictory**: inject logically conflicting criteria generated by GPT-4.1.
> - **50% removed**: randomly delete 50% of the original criteria, simulating rubrics with substantially incomplete coverage.
>
> For each noise setting, we train a Rubric-based RL baseline and RuscaRL on medical-domain data using Qwen2.5-7B-Instruct, and evaluate them on four medical benchmarks.
> Table R2 shows that RuscaRL is more robust to rubric noise: under mild perturbations (Ambiguous, Contradictory, 50% removed), it consistently outperforms Rubric-based RL, whereas under severe corruptions (Inverse, Negated) both methods degrade substantially.
> *These additional results have been updated in Appendix C.10 (Pages 25-26) of the revised manuscript.*
>
> Table R2. Robustness to rubric noise on medical benchmarks.
> |Rubric variant|HealthBench-500|LLMEval-Med|MedQA|MedMCQA|
> |-|-|-|-|-|
> |Initial Model|23.4±0.3|48.0±0.3|61.8±0.2|56.3±0.1|
> |+Rubric-based RL|||||
> |├─ Original|41.1±0.1|54.6±0.2|62.1±0.4|56.3±0.1|
> |├─ Inverse|7.1±0.2|41.3±0.5|61.3±0.1|55.8±0.1|
> |├─ Negated|2.9±0.5|36.8±1.4|60.5±0.0|55.6±0.1|
> |├─ Ambiguous|40.1±1.0|54.8±1.2|63.0±0.1|56.2±0.2|
> |├─ Contradictory|43.4±0.7|55.9±1.1|63.3±0.5|55.9±0.2|
> |└─ 50% removed|39.6±1.5|51.7±0.9|62.5±0.6|56.7±0.1|
> |+**RuscaRL (Ours)**|||||
> |├─ Original|**50.3±0.4**|**61.2±0.5**|**63.5±0.1**|**56.5±0.1**|
> |├─ Inverse|10.6±0.7|44.4±0.5|61.3±0.2|56.1±0.1|
> |├─ Negated|6.7±0.5|41.1±0.5|60.8±0.0|55.9±0.0|
> |├─ Ambiguous|46.2±0.3|59.6±1.4|62.2±0.2|56.0±0.1|
> |├─ Contradictory|45.7±0.7|56.8±0.6|63.2±0.5|56.2±0.1|
> |└─ 50% removed|44.6±0.2|54.9±0.6|63.2±0.2|56.3±0.1|
>
>
>
> **(2) Cross-domain rubric inconsistency.**
> In our original manuscript (Page 19), we have provided the cross-domain rubric experiments in Appendix D.2 (Mixed Training Analysis). For your convenience, we include again here the results from Table 5 in Appendix D.2 as Table R3.
> Mixed training with cross-domain rubrics still yields clear gains over the initial model on all benchmarks, even though it lags behind domain-specific training. This indicates that rubric inconsistency across domains does not severely harm RuscaRL and that the learned policy generalizes well.
>
> Table R3. Comparison of different training strategies.
> |Training Strategy|HealthBench-500|LLMEval-Med|MedQA|MedMCQA|WritingBench|Creative Writing|IFEVAL|IFBench|
> |-|-|-|-|-|-|-|-|-|
> |Initial|23.6|47.9|61.8|56.3|45.3|37.4|71.0|26.8|
> |Domain-specific|50.3|61.2|63.5|56.5|56.4|38.6|75.3|31.0|
> |Health-only|50.3|61.2|63.5|56.5|55.8|35.1|68.0|27.2|
> |Mixed Training|44.3|56.7|62.7|56.8|50.4|35.6|71.2|33.7|

---

> ### Author Response · Authors · 2025-11-21
> **Response (4/4)**
>
> **References**
>
> [1] HealthBench: Evaluating Large Language Models Towards Improved Human Health. arXiv 2025.
>
> [2] Kimi K2: Open Agentic Intelligence. arXiv 2025.
>
> [3] Baichuan-M2: Scaling Medical Capability with Large Verifier System. arXiv 2025.
>
> [4] DR Tulu: An open, end-to-end training recipe for long-form deep research. https://allenai.org/blog/dr-tulu 2025.
>
> [5] Rubrics as Rewards: Reinforcement Learning Beyond Verifiable Domains. arXiv 2025.
>
> [6] Reinforcement Learning with Rubric Anchors. arXiv 2025.
>
> [7] ResearchQA: Evaluating Scholarly Question Answering at Scale Across 75 Fields with Survey-Mined Questions and Rubrics. arXiv 2025.
>
> [8] ProfBench: Multi-Domain Rubrics Requiring Professional Knowledge to Answer and Judge. arXiv 2025.
>
> [9] ResearchRubrics: A Benchmark of Prompts and Rubrics For Evaluating Deep Research Agents. arXiv 2025.
>
> [10] Rubric-Based Benchmarking and Reinforcement Learning for Advancing LLM Instruction Following. arXiv 2025.

---

> ### Author Response · Authors · 2025-11-27
> **Looking Forward to your Reevaluation**
>
> Dear Reviewer CLiv,
>
> We are glad that the reviewer appreciates our attempt, and sincerely thank you for the constructive comments. As suggested, we have additionally provided detailed clarifications for our the focus of RuscaRL, and included further discussions and experiments on rubric noise resilience and computational cost. Please let us know if you have other questions or comments.
>
> Since the discussion window has less than a week remaining, we sincerely look forward to your reevaluation of our work and would very appreciate it if you could raise your score to boost our chance of more exposure to the community. Thank you very much!
>
> Best regards,
>
> Authors of RuscaRL

---

### Author Response · Authors · 2025-11-28
**Summary**

We thank all the reviewers for the constructive feedback!

In this post:

- (1) We summarize positive comments from the reviews.
- (2) We summarize the revisions of the manuscript.

In the individual replies, we address other comments.

(1) Positive comments

- **Two-Level Rubric-Based Scaffolding for Exploration**

  We propose a **rubric-based scaffolding mechanism**, inspired by instructional scaffolding, to address the “what cannot be explored cannot be learned” problem and exploration collapse in RL for general reasoning. It features two dynamic strategies:
   - **Intra-Group Scaffolding Differentiation**: Within each sampling group, samples receive different levels of rubric guidance, breaking homogeneity and expanding the exploration space for group-advantage RL.
   - **Inter-Step Scaffolding Decay**: To prevent reliance on rubric guidance, we gradually reduce the explicit criteria provided during training. This compels the model to internalize the rubric knowledge and transition towards independent reasoning.

  Reviewers agree that integrating instructional scaffolding with RL is novel and effective:
   - `[CLiv]`: "The integration of instructional scaffolding method into RL for LLMs is highly good."
   - `[FzkW]`: "The motivation (mitigating exploration bottlenecks in RL for LLMs) is interesting and relevant."
   - `[u4MF]`: "Motivation is clear and reasonable: the paper directly targets the exploration bottleneck ..."
   - `[nBsg]`: "Originality: The paper pioneers the introduction of instructional scaffolding theory ..."

- **Dual Roles of Rubrics: Guidance and Reward**

  Unlike prior approaches where rubrics functioned merely as rewards, RuscaRL leverages rubrics as both guidance and rewards:
  - **Rubric-based Scaffolding for Exploration**, providing checklist-style guidance during rollouts to expand the model’s reasoning space, enabling more diverse and novel trajectories.
  - **Rubric-based Reward for Exploitation**, providing stable, consistent reward estimation.

  Reviewers agree that this unified design enables a balance exploration–exploitation mechanism:
  - `[CLiv]`: "RuscaRL shows better sample efficiency (steeper Best-of-N curve) and avoids entropy collapse, evidencing better exploration control."
  - `[nBsg]`: "... RuscaRL achieves a better exploration-exploitation balance compared to baselines, avoiding premature entropy collapse or uncontrolled instability."


- **Extensive experiments and analyses**

  Rigorous ablation studies and analyses validate the key design choices and isolate the contribution of each component. Our empirical analyses further demonstrate that rubric-based scaffolding effectively improves sampling diversity and novelty:
   - `[FzkW]`: "Different ablation studies are included."
   - `[FzkW]`: "The results are overall interesting."
   - `[u4MF]`: "Insightful ablations on scaffolding ..."
   - `[nBsg]`: "The ablation studies rigorously validate the core design choices ..."

  These improvements translate into consistent performance gains across multiple models and benchmarks:
   - `[CLiv]`: "Demonstrates consistent gains across diverse benchmarks ..."
   - `[FzkW]`: "The empirical results show improvements across several domains and model sizes."
   - `[u4MF]`: "Strong empirical evidence: across multiple models and diverse tasks ..."
   - `[nBsg]`: "Quality: The robustness of the method is confirmed through extensive experimentation across a wide range of tasks and model scales ..."

(2) Revisions of the manuscript

- **Main Paper**

  * `[FzkW]`: (Figure 1 and Figure 2) Redesigned the figures.

  * `[FzkW, u4MF]`: (Section 2) Added related work with corresponding clarifications.

  * `[u4MF]`: (Section 5.1) Clarified the relationship between rubric-based RL and RuscaRL.

  * `[u4MF]`: (Section 5.2) Clarified that RuscaRL is trained solely on open-ended data.

- **Appendix**

  * `[CLiv，nBsg]`: (Appendix C.6) Added a more detailed analysis of training compute cost.

  * `[nBsg]`: (Appendix C.7) Added experiments for Direct Measurement of Scaffolding Internalization.

  * `[nBsg]`: (Appendix C.8) Added experiments on the Impact of Maximal Scaffolding on Novelty.

  * `[FzkW]`: (Appendix C.9) Included additional RL baselines.

  * `[CLiv, FzkW, nBsg]`: (Appendix C.10) Analyzed the framework’s robustness to noisy rubrics.

  * `[FzkW, u4MF]`: (Appendix E) Repeated experiments multiple times to ensure statistical robustness of experimental results.

  * `[u4MF]`: (Appendix F) Conducted experiments using high-temperature sampling.

  * `[FzkW]`: (Appendix G) Added a case study.


We have made every effort to address all of the weaknesses and questions raised by the reviewers, and we hope that our responses have satisfactorily resolved all of your concerns.

Best regards,

Authors of RuscaRL

---

### Meta-Review · Area_Chair_QXfK · 2026-01-07

**Summary:**

This work investigated the RL exploration issue, introducing the Rubric-Scaffolded Reinforcement Learning (RuscaRL) framework designed to overcome the exploration bottleneck in reinforcement learning (RL) for large language models (LLMs). The results show improvements across several domains and model sizes. The motivation (mitigating exploration bottlenecks in RL for LLMs) is interesting and relevant. However, most of the reviewers have concerns about the novelty of this work. Rubric rewards, scaffolded prompting, and gradual hint removal resemble prior work in rubric-guided RL, chain-of-thought prompting, and curriculum learning. Using rubric-guided multi-sample rollouts with LLM-as-a-Judge evaluations can also be computationally expensive.

**Reviewer Concerns:**

Three of the reviewers have concerns about the novelty of this work. The novelty needs to be further enhanced.

**Reviewer Scores:**

Reviewer FzkW may increase their score. Other reviewers may keep the rating.

---

### Decision · Program_Chairs · 2026-01-26

Reject